# Pixelwise Gradient Model with GAN for Virtual Contrast Enhancement in MRI Imaging

**DOI:** 10.3390/cancers16050999

**Published:** 2024-02-29

**Authors:** Ka-Hei Cheng, Wen Li, Francis Kar-Ho Lee, Tian Li, Jing Cai

**Affiliations:** 1Department of Health Technology and Informatics, The Hong Kong Polytechnic University, Hong Kong SAR, China; ka-hei.cheng@connect.polyu.hk (K.-H.C.); polyu-wen.li@polyu.edu.hk (W.L.); litian.li@polyu.edu.hk (T.L.); 2Department of Clinical Oncology, Queen Elizabeth Hospital, Hong Kong SAR, China; leekh4@ha.org.hk; 3The Hong Kong Polytechnic University Shenzhen Research Institute, Shenzhen 518000, China

**Keywords:** virtual contrast enhancement, tumor contrast, MR-guided radiotherapy, nasopharyngeal carcinoma

## Abstract

**Simple Summary:**

This paper presents a novel approach to produce virtual contrast enhanced (VCE) images for nasopharyngeal cancer (NPC) without the use of contrast agents, which carry certain risks. This model uses pixelwise gradient term to capture the shape and a GAN terms to capture the texture of the real contrast enhanced T1C images With similar accuracy to existing models, our method shows an advantage in reproducing texture closer to the realistic contrast-enhanced images. This results are tested by various measures, including mean absolute error (MAE), mean square error (MSE) and structural similarity index (SSIM) for similarity accuracy; total mean square variation per mean intensity (TMSVPMI), the total absolute vari-ation per mean intensity (TAVPMI), Tenengrad function per mean intensity (TFPMI) and variance function per mean intensity (VFPMI) Various variations of the model, including fine-tuning of the hyperparameters, normalization methods on the images and using single modality, have also been investigated to test the optimal performance.

**Abstract:**

**Background**: The development of advanced computational models for medical imaging is crucial for improving diagnostic accuracy in healthcare. This paper introduces a novel approach for virtual contrast enhancement (VCE) in magnetic resonance imaging (MRI), particularly focusing on nasopharyngeal cancer (NPC). **Methods**: The proposed model, Pixelwise Gradient Model with GAN for Virtual Contrast Enhancement (PGMGVCE), makes use of pixelwise gradient methods with Generative Adversarial Networks (GANs) to enhance T1-weighted (T1-w) and T2-weighted (T2-w) MRI images. This approach combines the benefits of both modalities to simulate the effects of gadolinium-based contrast agents, thereby reducing associated risks. Various modifications of PGMGVCE, including changing hyperparameters, using normalization methods (z-score, Sigmoid and Tanh) and training the model with T1-w or T2-w images only, were tested to optimize the model’s performance. **Results**: PGMGVCE demonstrated a similar accuracy to the existing model in terms of mean absolute error (MAE) (8.56 ± 0.45 for Li’s model; 8.72 ± 0.48 for PGMGVCE), mean square error (MSE) (12.43 ± 0.67 for Li’s model; 12.81 ± 0.73 for PGMGVCE) and structural similarity index (SSIM) (0.71 ± 0.08 for Li’s model; 0.73 ± 0.12 for PGMGVCE). However, it showed improvements in texture representation, as indicated by total mean square variation per mean intensity (TMSVPMI) (0.124 ± 0.022 for ground truth; 0.079 ± 0.024 for Li’s model; 0.120 ± 0.027 for PGMGVCE), total absolute variation per mean intensity (TAVPMI) (0.159 ± 0.031 for ground truth; 0.100 ± 0.032 for Li’s model; 0.153 ± 0.029 for PGMGVCE), Tenengrad function per mean intensity (TFPMI) (1.222 ± 0.241 for ground truth; 0.981 ± 0.213 for Li’s model; 1.194 ± 0.223 for PGMGVCE) and variance function per mean intensity (VFPMI) (0.0811 ± 0.005 for ground truth; 0.0667 ± 0.006 for Li’s model; 0.0761 ± 0.006 for PGMGVCE). **Conclusions**: PGMGVCE presents an innovative and safe approach to VCE in MRI, demonstrating the power of deep learning in enhancing medical imaging. This model paves the way for more accurate and risk-free diagnostic tools in medical imaging.

## 1. Introduction

In the rapidly advancing field of medical imaging, the refinement of sophisticated models for image analysis plays a pivotal role in boosting diagnostic precision and patient care. Magnetic resonance imaging (MRI), among various imaging techniques, demonstrates its prowess in delivering high-resolution imagery of soft tissues, notably without employing ionizing radiation. T1-weighted (T1-w) and T2-weighted (T2-w) images, in particular, stand as cornerstones in the diagnosis of diverse medical conditions [1,2].

Virtual-contrast-enhanced T1 images (VCE T1C images) represent a novel technique designed to emulate the visibility of certain tissues and fluids typically accentuated by contrast agents. In clinical practice, these agents are frequently utilized to amplify the distinction between normal and abnormal tissues, especially in the brain, aiding in the delineation of tumors, inflammation and other pathologies. For instance, in cases of nasopharyngeal cancer (NPC), gadolinium-based contrast agents are administered to enhance tumor visibility. However, the usage of these agents is not without associated risks [3,4,5,6,7,8,9,10,11,12,13]. Virtual contrast enhancement, thus, emerges as a safer alternative, mimicking the effects of contrast agents through deep learning (DL) applications. The incorporation of DL in image synthesis has recently attracted considerable attention in the field of medical imaging. Its potential in discerning complex tumor characteristics [14,15,16] has spurred research into the synthesis of virtual-contrast-enhanced MRI (VCE-MRI) from non-contrast scans, particularly for brain cancer patients.

The development of VCE-MRI involves training DL algorithms with large datasets of MRI scans, both with and without contrast agents. By learning the patterns and characteristics of gadolinium-enhanced images, these algorithms can generate virtual-contrast-enhanced images from standard T1-w and T2-w scans. This process not only obviates the need for contrast agents but also has the potential to reduce scan time and costs associated with the use of these agents.

The medical field has recently spotlighted the advancement of DL in generating synthesized images [17,18,19,20,21,22,23]. Deep neural networks’ ability to dissect and understand the intricate details of tumor characteristics has led to the innovation of creating VCE-MRI images from non-contrast MRI scans for brain cancer patients [7,24]. Specifically, Gong et al. utilized a U-shaped DL model to merge MRI images without a gadolinium-based contrast agent and with a low dose of a gadolinium-based contrast agent, achieving VCE-MRI images that mimic those produced with a full dose of a gadolinium-based contrast agent. This study showcased the potential of DL in extracting contrast enhancement details from full-dose gadolinium-based contrast agent MRI images and generating vceT1w MRI images of satisfactory quality. Building on this groundwork, a three-dimensional Bayesian neural network that integrates ten different MRI techniques to produce VCE-MRI images was introduced [7]. This confirmed the DL network’s capacity to utilize various non-contrast imaging methods for synthesizing images. Despite these encouraging results, the current DL models face challenges in harnessing the full potential of the diverse information available from different imaging inputs. This limitation’s impact becomes more significant when diagnosing deeply infiltrative NPC, due to the complex interaction of pixel intensity across different imaging modalities [25].

Our model, Pixelwise Gradient Model with GAN for Virtual Contrast Enhancement (PGMGVCE), employs pixelwise gradient methods to delineate the shape of VCE images, complemented by the use of a Generative Adversarial Network (GAN) to replicate the image contrast of VCE. Pixelwise gradient originated from image registration [26,27,28], and, to our knowledge, our work is the first to apply it to image synthesis, in particular, VCE-MRI. The evaluation of our models encompasses not only quantitative accuracy metrics, such as mean absolute error (MAE), mean square error (MSE) and structural similarity (SSIM), but also qualitative assessments of texture. It was observed that the VCE images generated using the model in [22] exhibit excessively smooth textures compared to actual T1C images. The novelty of PGMGVCE, based on pixelwise gradient techniques, is that it demonstrates a texture in VCE images more akin to realistic T1C images. This is evidenced by various metrics introduced in this paper, such as the total mean square variation per mean intensity (TMSVPMI), total absolute variation per mean intensity (TAVPMI), Tenengrad function per mean intensity (TFPMI) and variance function per mean intensity (VFPMI). Despite similar mean absolute errors between images produced by PGMGVCE and the model in [22] when compared with ground truth, the improved textural fidelity of PGMGVCE images suggests its superiority over the model in [22].

Section 2.1 and Section 2.2 introduce the model architectures of the PGMGVCE model. Section 2.3 introduces methods to evaluate the performance of the models. Section 2.4 discusses the data preprocessing steps. Section 3.1 shows the results of the VCE images. Section 2.5 and Section 3.2 include comprehensive comparisons of various adaptations of the PGMGVCE model. These comparisons encompass modifications in hyperparameters, the application of different image normalization techniques and training the models exclusively with either T1-w or T2-w images, thereby enriching the study with a thorough analytical perspective. Section 4 and Section 5 are the discussion and conclusions, respectively.

## 2. Methods

### 2.1. Model Architecture of Pixelwise Gradient Model with GAN for Virtual Contrast Enhancement (PGMGVCE)

The PGMGVCE architecture is depicted in Figure 1. This model is trained using the pixelwise gradient method in conjunction with a Generative Adversarial Network (GAN), enabling it to accurately replicate the shape and contrast of input images. The gradient method, initially developed for image registration [26,27,28] in the context of image registration, was adapted for VCE. This adaptation begins by calculating the gradient of an image:(1)$$\nabla x\left(i,j\right)=\left(x\left(i+1,j\right)-x\left(i,j\right),x\left(i,j+1\right)-x\left(i,j\right)\right)$$
where x(i,j) represents the (i,j) pixel of image. To capture the shape of the input images, we consider normalized gradients $n=\frac{\nabla x\left(i,j\right)}{\|\nabla x\left(i,j\right)\|+\epsilon }$, where ϵ is a small constant to avoid division of zero. If the output images capture the same shape as the input images, the gradient will point to the same or opposite directions, as the gradient of pixel intensity is a geometric quantity. The alignment of the gradient with the input images’ shape is measured by the square of the dot product between the input and output normalized gradients, forming the basis of our loss function.
(2)$$Pixelwise\;Gradient(y_{output},z_{ground\;truth})=-\sum_{pixel}{\left(n_{output}\cdot n_{ground\;truth}\right)}^{2}$$
where $y_{output},z_{ground\;truth}$ are the output and ground-truth real T1C images, respectively. 

To learn the image contrast, we employ GAN loss:(3)$$GAN\;Loss=GAN\left(output\;image,ground\;truth\;T1C\;image\right)$$
utilizing a Least Squares Generative Adversarial Network (LSGAN).

### 2.2. Implementation of the PGMGVCE

The model accepts T1-weighted and T2-weighted MRI slices as input. A series of convolutional layers are used for initial feature extraction. These layers progressively downsample the image while increasing the feature map depth. Then, a module is designed to effectively integrate information from T1-w and T2-w images. It extracts and combines features from each modality, leveraging their complementary nature to enhance contrast and detail. Then, the model employs a set of trainable weights that adjust dynamically during training, optimizing the contribution of each modality based on the fusion target. This guides the fusion process by evaluating and weighting the importance of features from different modalities. This system ensures that the most relevant features for contrast enhancement are prioritized. Then, the output is forwarded to a module that utilizes separate convolutional pathways for each modality, followed by feature fusion layers that intelligently merge the extracted features. This approach ensures that the unique characteristics of each modality are preserved and effectively utilized. This module enhances the model’s ability to focus on salient features within the medical images, such as areas indicating pathological changes.

After that is the discriminator, which incorporates a patch-based approach, where it evaluates the authenticity of different regions of the image separately. This localized assessment enables a more detailed and accurate evaluation of the image quality. This is to improve the realism and diagnostic quality of the fused images.

Each of these components is designed to work in synergy, with the outputs of one module feeding into or influencing others. This integration ensures a cohesive and effective processing pipeline, resulting in high-quality, contrast-enhanced medical images.

The model is trained using an Adam optimizer with a learning rate of 0.0002 and a beta1 value of 0.5. Training is conducted in mini-batches, with each batch containing an equal mix of T1-weighted and T2-weighted images. To ensure stable training, a gradient penalty is applied to the discriminator, encouraging smoother gradients in the generated images. We performed 14,000 iterations for training.

### 2.3. Evaluation of the Models

The accuracy of the PGMGVCE and the model in [22] are evaluated using mean absolute error (MAE), mean square error (MSE) and structural similarity index (SSIM). The metrics are expressed as follows:(4)$$AE=\frac{1}{N}\left|y\left(x\right)-g(x)\right|$$
(5)$$MSE=\frac{1}{N}{\left(y\left(x\right)-g(x)\right)}^{2}$$
(6)$$SSIM=\frac{\left(2\mu_{y\left(x\right)}\mu_{g\left(x\right)}+c_{1})(2\sigma_{y\left(x\right)g\left(x\right)}+c_{2}\right)}{\left({\mu_{y\left(x\right)}}^{2}+{\mu_{g\left(x\right)}}^{2}+c_{1})({\sigma_{y\left(x\right)}}^{2}+{\sigma_{g\left(x\right)}}^{2}+c_{2}\right)}$$
where N is the number of pixels in each image slice, and $y\left(x\right)$ and g(x) denote the synthetic VCE T1C images and the ground truth, respectively. $\mu_{y\left(x\right)}$, $\mu_{g\left(x\right)}$ and $\sigma_{y\left(x\right)}$, $\sigma_{g\left(x\right)}$ are the means and variances of the synthetic images and the ground truth, whereas $\sigma_{y\left(x\right)g\left(x\right)}$ is the covariance of $y\left(x\right)$ and g(x). $c_{1}={\left(k_{1}L\right)}^{2}$ and $c_{2}={\left(k_{2}L\right)}^{2}$ are 2 variables used to stabilize the division by the weak denominator, and L is the dynamic range of the pixel values. Here, L=4095, $k_{1}=0.01$ and $k_{2}=0.03$ were set by default.

To quantify the smoothness of VCE T1C images, four metrics are introduced. It can be used as a measure to check the difference in texture of the images. All metrics divide the mean intensity since relative pixel intensity variations capture the texture of the images. For example, if an image is multiplied by a constant, the total variations will also be multiplied by that constant, but the texture of the image should be invariant under that multiplication, as it is intrinsic to the image.

The first two metrics are the total mean square variation per mean intensity (TMSVPMI) and the total absolute variation per mean intensity (TAVPMI), which are, respectively, defined as
(7)$$TMSVPMI=\frac{1}{\mu }\frac{1}{Number\;of\;pixels}\sum_{pixels}\sqrt{{\left(\nabla_{x}image\right)}^{2}+{\left(\nabla_{y}image\right)}^{2}}$$
(8)$$TMAVPMI=\frac{1}{\mu }\frac{1}{Number\;of\;pixels}\sum_{pixels}\left(\left|\nabla_{x}image\right|+\left|\nabla_{y}image\right|\right)$$
where μ is the mean intensity of the image.

The third metric is the Tenengrad function per mean intensity (TFPMI) which is based on [29]:(9)$$TFPMI=\frac{1}{\mu }\frac{1}{Number\;of\;pixels}\sum_{pixels}\sqrt{G_{x}^{2}+G_{y}^{2}}$$
where μ is the mean intensity of the image, and $G_{x}$ and $G_{y}$ are the Sobel operators which are, respectively, the convolutions of the images with the kernels:$$\left(\begin{array}{ccc} -1 & 0 & 1\\ -2 & 0 & 2\\ -1 & 0 & 1 \end{array}\right)\;\mathrm{and}\;\left(\begin{array}{ccc} 1 & 2 & 1\\ 0 & 0 & 0\\ -1 & -2 & -1 \end{array}\right).$$

The fourth metric, which is also motivated by [29], is the variance function per mean intensity (VFPMI):(10)$$VFPMI=\frac{1}{\mu }\frac{1}{Number\;of\;pixels}\sum_{pixels(x,y)}{\left(I\left(x,y\right)-\mu \right)}^{2}$$
where I is the image, and μ is the mean intensity of the image.

The smaller the indices, the smoother the image.

### 2.4. Data Preprocessing

The dataset, approved by the Research Ethics Committee in Hong Kong (reference number: UW21-412), consisted of 80 NPC patients at stages I to IVb, imaged with T1-w, T2-w and T1C MRI using a 3T-Siemens scanner with TR: 620 ms, TE: 9.8 ms; TR: 2500 ms, TE: 74 ms; and TR: 3.42 ms, TE: 1.11 ms, respectively. The average age of the patients was 57.6 ± 8.6, including 46 males and 34 females. Each patient’s 3D image was converted into 2D slices for training. Image alignment was necessary for contrast enhancement; thus, 3D T1-w and T1C MR images were registered with T2-w MR images using 3dSlicer [30] with a B-Spline transform and mutual information as the similarity metric. Then, different 2D slices were extracted from the 3D images. The dataset was randomly divided into 70 patients for training and 10 for testing, resulting in 3051 and 449 2D image slices for each modality, respectively. Figure 2 shows some sample processed T1-w and T2-w images. These slices were resized to 192 × 192, a dimension compatible with the convolutional network structure.

### 2.5. Different Variations of the PGMGVCE 

#### 2.5.1. Fine-Tuning the Hyperparameter

We tested various hyperparameter values between pixelwise gradient terms and GAN terms to evaluate the performance. Ratios of 10:1, 1:1 and 1:10 are discussed here.

#### 2.5.2. Different Normalization Methods on Images

Normalization is a crucial preprocessing step to standardize the range of pixel intensity values, thereby enhancing the model’s ability to learn and generalize from the data. Different datasets may have different distributions. Some might be Gaussian, while others might have a uniform or skewed distribution. Each normalization method is tailored to work best with a specific type of distribution. Moreover, using the appropriate normalization for a particular data distribution can make the model more robust to variations and outliers, thereby improving its performance and accuracy. Furthermore, when combining features with different scales (e.g., T1-weighted and T2-weighted MRI images), normalization ensures that each feature contributes equally to the analysis and is not dominated by those on larger scales. Normalization can help in emphasizing the importance of smaller-scale features that might be critical for diagnosis in medical images. 

Different normalization methods (z-score, Sigmoid and Tanh) [31] are also applied to the images, where
(11)$$x_{z-score}=\frac{x-\mu_{x}}{\delta_{x}}$$
(12)$$x_{sigmoid}=\frac{1}{1+e^{-\frac{x-\mu_{x}}{\delta_{x}}}}$$
(13)$$x_{tanh}=\frac{1}{2}\left[\tanh\left(0.01(\frac{x-\mu_{x}}{\delta_{x}}\right)+1\right]$$
where x represents the intensities of each patient volume, and $\mu_{x}$ and $\delta_{x}$ are the mean value and standard deviation of the patient. $x_{z\_score}$, $x_{sigmoid}$ and $x_{tanh}$ represent the corresponding values of patient data after z-normalization, Sigmoid and Tanh normalization methods, respectively.

Z-score normalization, or standard score normalization, involves rescaling the data to have a mean of 0 and a standard deviation of 1. Z-score normalization ensures that each feature contributes equally to the analysis, which is critical when combining features of different scales and units. It enhances the model’s sensitivity to outliers, which can be vital for identifying anomalies in medical images. When the underlying data distribution is Gaussian, z-score normalization makes the features more Gaussian-like, which is an assumption in many machine learning models.

Sigmoid normalization transforms data using the Sigmoid function to constrain values within a range of 0 to 1. It bounds the input into a fixed range, which can be beneficial for models that are sensitive to input scale and distribution. The smooth nature of the Sigmoid function provides smooth gradients, which can aid in the convergence during the training of deep learning models. In medical images, this can help preserve the context and relative contrast between different tissue types while standardizing the overall intensity scale.

Tanh normalization is similar to Sigmoid but rescales the data to a range between 0 and 1. Data are centered around 1/2, which can lead to better performance in models where the sign of the data is important. The steeper gradient of Tanh (compared to Sigmoid) around the center can lead to faster learning and convergence in some cases. For medical images, this method can enhance contrast between areas of interest, potentially improving the model’s ability to learn and distinguish pathological features.

For the Sigmoid and Tanh normalization, we first transform the image intensity according to (11)–(13) to all the T1-w, T2-w and T1C images. Then, we train and apply the deep learning models to the transformed images. Finally, we perform inverse Sigmoid and inverse Tanh, respectively, to the output synthetic images to obtain images comparable to those of the z-score normalization.

#### 2.5.3. Using Single Modality for Contrast Enhancement

We assessed the performance of using both T1-w and T2-w images versus a single modality for contrast enhancement. The mean absolute error ratio (MAER), mean square error ratio (MSER) and structural similarity ratio (SSIMR) were computed as defined by
(14)$$MAER=\frac{{MAE}_{T1/T2}-{MAE}_{T1\;and\;T2}}{{MAE}_{T1\;and\;T2}}$$
(15)$$MSER=\frac{{MSE}_{T1/T2}-{MSE}_{T1\;and\;T2}}{{MSE}_{T1\;and\;T2}}$$
(16)$$SSIMR=\frac{{SSIM}_{T1/T2}-{SSIM}_{T1\;and\;T2}}{{SSIM}_{T1\;and\;T2}}$$
where ${MAE}_{T1/T2}$, ${MSE}_{T1/T2}$ and ${SSIM}_{T1/T2}$ are, respectively, the MAE, MSE and SSIM between the ground-truth VCE images and the image output of T1/T2 only. ${MAE}_{T1\;and\;T2}$, ${MSE}_{T1\;and\;T2}$ and ${SSIM}_{T1\;and\;T2}$ are, respectively, the MAE, MSE and SSIM between the ground-truth VCE images and the image output of both T1 and T2, comparing the performance of combined modality inputs against single-modality inputs.

## 3. Results

### 3.1. Comparison between the Model in [22] and the PGMGVCE

We compared the PGMGVCE with the model in (16) using MAE, MSE and SSIM (Table 1). The comparison statistics between the models and the ground truth are very close to each other. As shown in Figure 3, the VCE images qualitatively also look very similar. These indicate that, in terms of accuracy, the PGMGVCE is similar to that in [22]. However, qualitatively, the texture of the model in [22] appears to be smoother than the ground-truth T1C. This can be illustrated by the texture statistics. The PGMGVCE has a texture closer to the ground truth than the model in [22]. The TMSVPMI, TAVPMI, TFPMI and VFPMI of the ground-truth T1C, VCE images produced by the PGMGVCE and model in [22] are shown in Table 2. The p-values of TMSVPMI, TAVPMI, TFPMI and VFPMI of the model by [22] are the same as the ground-truth T1C, that is, 0.002, 0.003, 0.004 and 0.0001, respectively, which is statistically significant to claim that the TMSVPMI, TAVPMI, TFPMI and VFPMI of the PGMGVCE are larger than those of the model in [22]. This is an indication that the texture of the result of the PGMGVCE is closer to the realistic ones.

### 3.2. Comparison of Different Variations of the PGMGVCE

#### 3.2.1. Fine-Tuning the Hyperparameter

Figure 4 shows sample images trained with the ratio of the hyperparameter for the pixelwise gradient loss term and GAN loss term of 10:1, 1:1 and 1:10. The MAE and MSE between the images were synthesized with the ratio of pixelwise gradient loss to GAN loss of 10:1 and the ground truth of 10.517 ± 0.247 and 14.008 ± 0.284, respectively; the ratio of pixelwise gradient loss to GAN loss of 1:1 and the ground truth of 9.904 ± 0.178 and 12.866 ± 0.197, respectively; and the ratio of pixelwise gradient loss to GAN loss of 1:10 and the ground truth of 10.517 ± 0.231 and 13.811 ± 0.273, respectively. It demonstrated that the ratio of 1:1 between the pixelwise gradient loss and GAN loss shows slightly better performance than the other two ratios.

#### 3.2.2. Different Normalization Methods on Images

Figure 5 shows sample images with z-score, Sigmoid and Tanh normalization methods before training. From qualitative inspection, the differences between different normalization methods do not deviate much from each other and the ground truth. For detailed statistics, the MAE and MSE between the image synthesized with z-score normalization and the ground truth are 4.222 ± 0.843 and 4.662 ± 0.918, respectively; the MAE and MSE between the image synthesized with Sigmoid normalization and the ground truth are 3.814 ± 0.612 and 4.213 ± 0.726, respectively; and the MAE and MSE between the image synthesized with Tanh normalization and the ground truth are 3.932 ± 0.672 and 4.175 ± 0.892, respectively. This shows that Sigmoid normalization slightly outperforms the other two.

#### 3.2.3. Using Single Modality for Contrast Enhancement

Figure 6 shows the results of synthesizing VCE images using T1-w, T2-w and both T1-w and T2-w images. The MAER of using T1-w images only is 0.456 ± 0.102 while the MAER of using T2-w images only is 0.389 ± 0.098. The MSER of using T1-w only is 0.447 ± 0.146 while the MSER of using T2-w images only is 0.413 ± 0.161. The SSIMR of using T1-w only is 0.283 ± 0.103 while the SSIMR of using T2-w images only is 0.241 ± 0.112.

## 4. Discussion

Our model’s architecture incorporates convolutional layers for initial feature extraction, succeeded by modules that integrate and prioritize features from each imaging modality. Employing a blend of pixelwise gradient methods and GANs, our model captures intricate details from the input images. The gradient method, inspired by image registration techniques [26,27,28], is adept at detecting subtle variations in the shape and texture characteristics of different tissues and pathologies. By calculating and normalizing the image gradient, the model discerns the geometric structure of tissues, aiding in the high-fidelity reconstruction of enhanced images [32]. GANs, recognized for their capacity to generate lifelike images, are applied here to ensure that the synthesized T1C images are not only structurally precise but visually indistinguishable from actual contrast-enhanced scans. The dynamic interaction between the discriminative and generative elements of GANs compels the model to yield results that fulfill the stringent criteria necessary for clinical diagnosis [33].

Comparative analysis with the model in [22] reveals that, while basic accuracy metrics (MAE, MSE, SSIM) are comparable, the PGMGVCE demonstrates superior texture representation. The metrics of TMSVPMI, TAVPMI, TFPMI and VFPMI for the model in [22] are significantly lower than the ground-truth T1C, indicating that its results are overly smooth and may lack critical detail. In contrast, these metrics for the PGMGVCE closely match those of the ground-truth T1C, suggesting a more realistic texture replication. This could be attributed to the PGMGVCE incorporating pixelwise gradient in its loss term, enhancing its ability to capture the authentic texture of T1C images.

Various iterations of the PGMGVCE were examined, particularly the impact of different hyperparameter ratios between pixelwise gradient and GAN components. After extensive trial and error, a 1:1 ratio was identified as optimal. Regarding image normalization methods, Sigmoid normalization was found to be superior, followed by Tanh and z-score normalization. When considering the use of single modalities for VCE image synthesis, it is evident that using both T1-w and T2-w images yields better results than using either modality alone, as the latter only captures partial anatomical information. This conclusion is supported by higher MAER, MSER and SSIMR values when using single modalities.

There are some limitations of our study. The model’s performance heavily relies on the quality and diversity of the training data. Additionally, incorporating other MRI modalities or sequences might further amplify the model’s diagnostic capabilities. Future investigations should focus on enhancing the model’s generalizability by training it with a more varied dataset and could also explore the real-time application of this model in clinical settings to assess its practicality and effectiveness in routine clinical workflows.

## 5. Conclusions

This study introduces a novel method for VCE in MRI imaging through a deep learning model that effectively combines pixelwise gradient methods with GANs. Our model excels in utilizing the complementary strengths of T1-w and T2-w MRI images, thereby synthesizing T1C images that are visually akin to actual contrast-enhanced scans. The fusion of these imaging modalities is key, as it captures a more exhaustive representation of the anatomy and pathology, thus increasing the diagnostic utility of the images.

In summary, this study presents an innovative approach to virtual contrast enhancement in MRI imaging, leveraging deep learning to reduce the risks associated with contrast agents in VCE images. The ability of our model, the PGMGVCE, to generate images with authentic textures and its potential to offer safer and more precise diagnostics, represents a significant advancement in medical imaging. While the PGMGVCE demonstrates comparable accuracy to the existing model (16), its enhanced texture replication sets it apart, underlining its advantage in realistically capturing VCE images.

The clinical implications of our study are noteworthy. By offering a safer alternative to gadolinium-based contrast agents, the PGMGVCE may diminish the risks linked with contrast-enhanced MRI scans. The improved texture accuracy of the synthesized images could potentially lead to enhanced diagnosis and patient management, particularly in the detection and characterization of NPC.

A limiting aspect of our synthesis network’s efficacy is its training solely on T1- and T2-weighted MRI images. It appears these types of images may not encapsulate all the necessary details for effective contrast synthesis. This issue could potentially be mitigated by incorporating additional MRI techniques (like diffusion-weighted MRI) into our network’s input. Furthermore, the model only underwent evaluation using a single dataset. Therefore, its performance and ability to generalize across different scenarios require further examination in subsequent research.

One possible future direction would be performing segmentation on the tumor region to evaluate the performance of the tumor enhancement. Different VCE methods and the ground-truth real T1C can be segmented to compare with each other. This would be a method to test if the tumor contrast of the VCE images is enhanced.

## Figures and Tables

**Figure 1 cancers-16-00999-f001:**
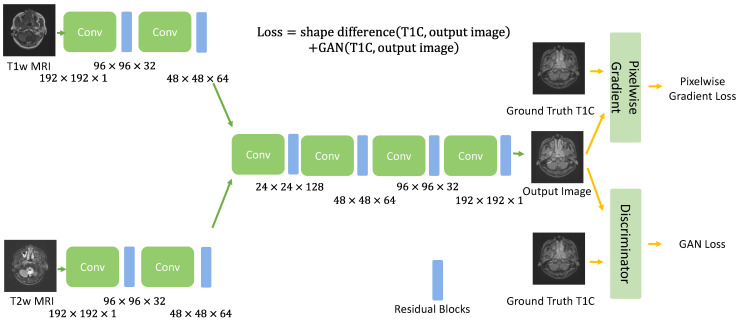
Architecture of the deep learning model for the proposed model, Pixelwise Gradient Model with GAN for Virtual Contrast Enhancement (PGMGVCE). The multiplications in the figure indicate the dimension and number of channels in that layer.

**Figure 2 cancers-16-00999-f002:**
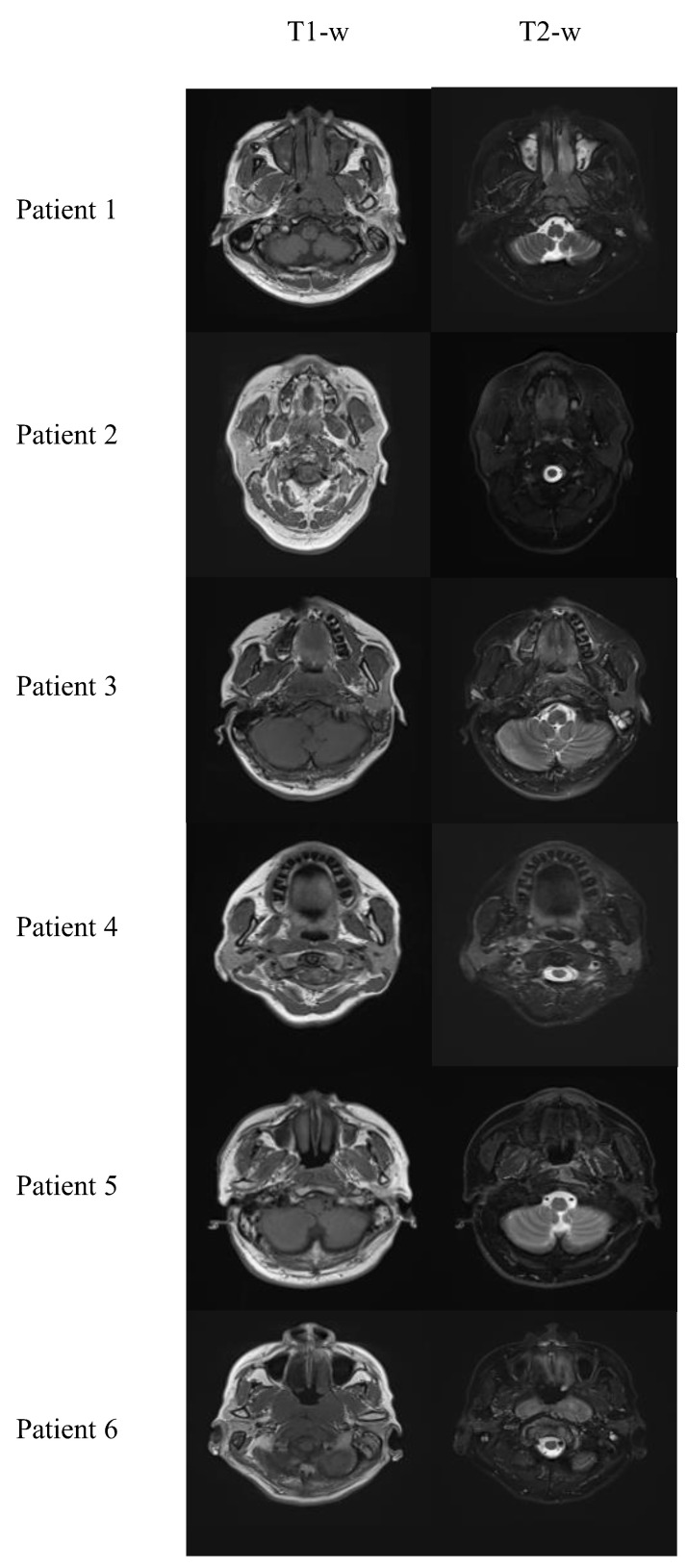
Sample processed T1-w and T2-w images.

**Figure 3 cancers-16-00999-f003:**
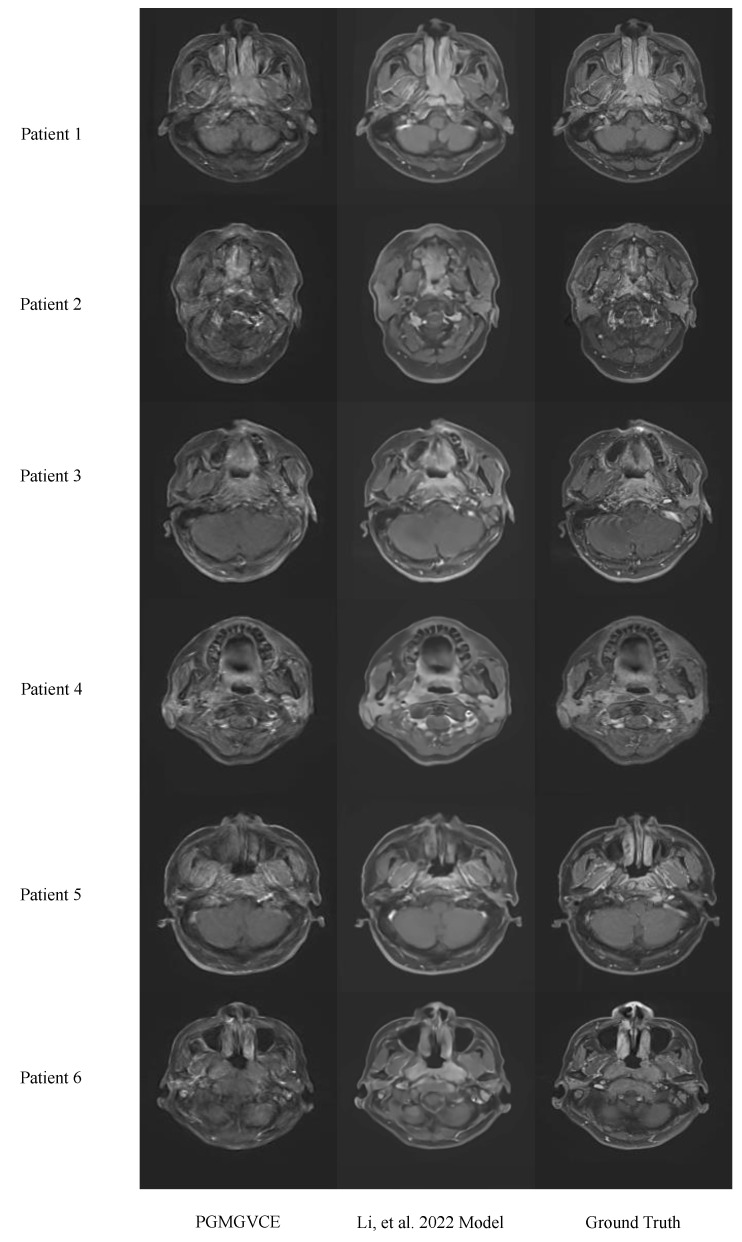
Results of the Pixelwise Gradient Model with GAN for Virtual Contrast Enhancement (PGMGVCE), the model in [22] and ground truth. It can be seen, qualitatively, that the texture of model in [22] appears to be smoother than the ground truth.

**Figure 4 cancers-16-00999-f004:**
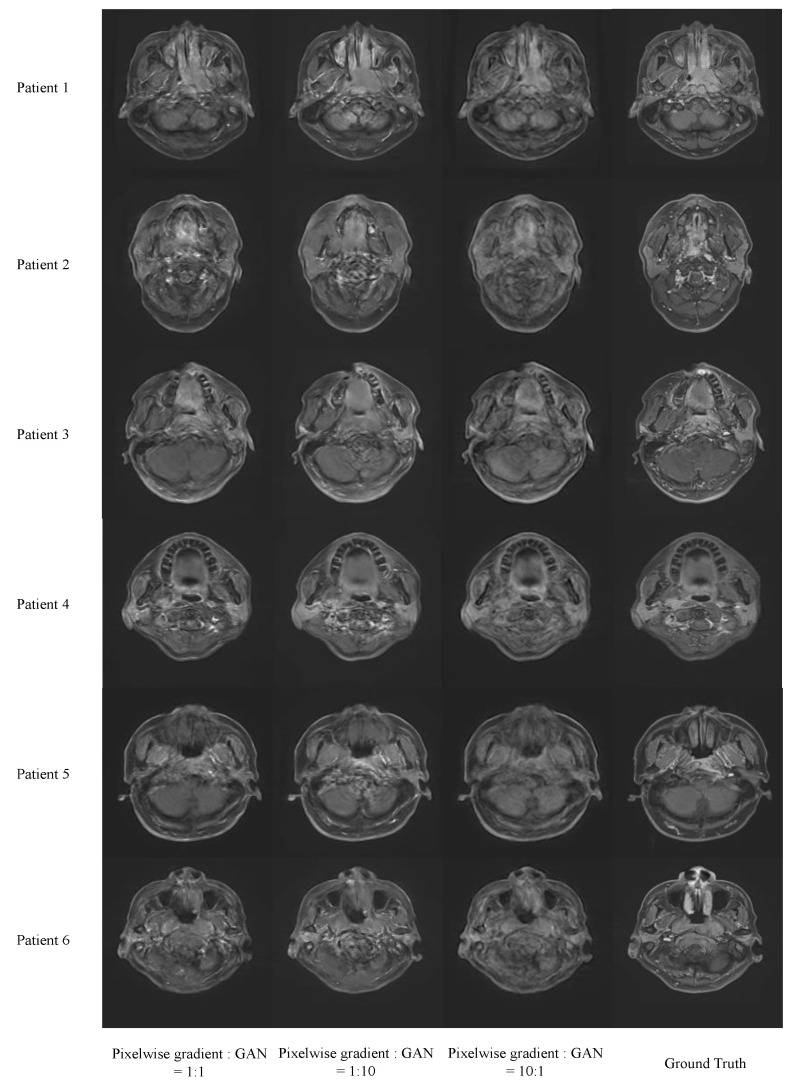
Results of the Pixelwise Gradient Model with GAN for Virtual Contrast Enhancement (PGMGVCE) with different ratios of hyperparameter of the pixelwise gradient term and the GAN term.

**Figure 5 cancers-16-00999-f005:**
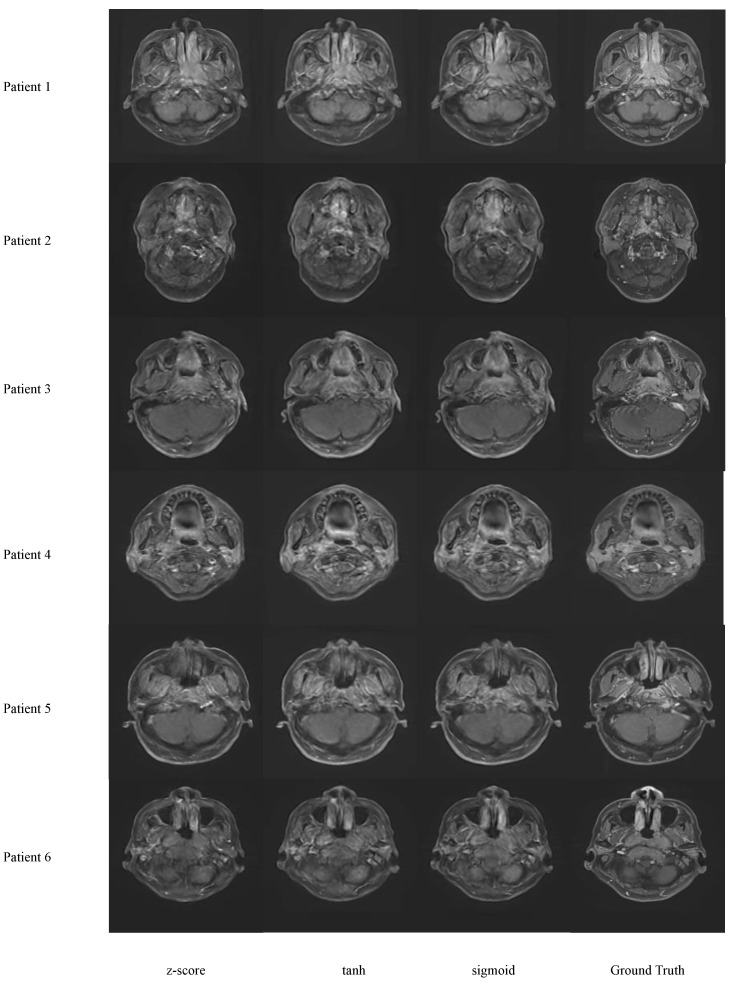
Results of the Pixelwise Gradient Model with GAN for Virtual Contrast Enhancement (PGMGVCE) with different normalization methods of the images.

**Figure 6 cancers-16-00999-f006:**
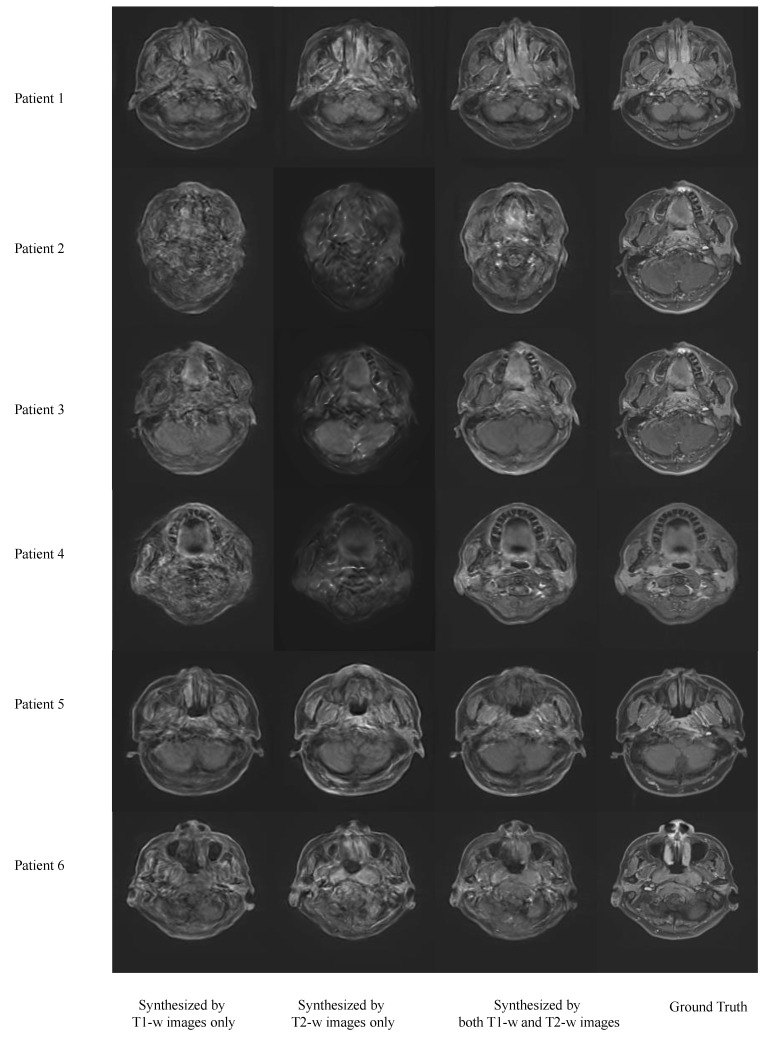
Results of virtual contrast enhanced (VCE) images synthesized using T1-w or T2-w images only and both T1-w and T2-w images.

**Table 1 cancers-16-00999-t001:** Mean absolute error (MAE), mean square error (MSE) and structural similarity index (SSIM) between the ground-truth real-contrast-enhanced T1C and virtual contrast enhanced (VCE) images produced by the Pixelwise Gradient Model with GAN for Virtual Contrast Enhancement (PGMGVCE) and model in [22].

	Ground-Truth and VCE Images Produced by the PGMGVCE	Ground Truth and Model in [22]
MAE	8.56 ± 0.45	8.72 ± 0.48
MSE	12.43 ± 0.67	12.81 ± 0.73
SSIM	0.71 ± 0.08	0.73 ± 0.12

**Table 2 cancers-16-00999-t002:** The total mean square variation per mean intensity (TMSVPMI), total absolute variation per mean intensity (TAVPMI), Tenengrad function per mean intensity (TFPMI) and variance function per mean intensity (VFPMI) of the ground-truth real-contrast-enhanced T1C and virtual contrast enhanced (VCE) images produced by the Pixelwise Gradient Model with GAN for Virtual Contrast Enhancement (PGMGVCE) and model in [22].

	Ground Truth	Model in [22]	PGMGVCE
TMSVPMI	0.124 ± 0.022	0.079 ± 0.024	0.120 ± 0.027
TAVPMI	0.159 ± 0.031	0.100 ± 0.032	0.153 ± 0.029
TFPMI	1.222 ± 0.241	0.981 ± 0.213	1.194 ± 0.223
VFPMI	0.0811 ± 0.005	0.0667 ± 0.006	0.0761 ± 0.006

## Data Availability

For original data and computer programs, please contact the first author (K.-H.C.) (khcheng9209@gmail.com).

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
