# Peer review of "Pixelwise Gradient Model with GAN for Virtual Contrast Enhancement in MRI Imaging"

_cancers, 2024, doi:10.3390/cancers16050999_

Round 1
Reviewer 1 Report
Comments and Suggestions for Authors
Please address all my comments listed in the attached file.

Moderate editing of English language required. There are many examples of noun-verb inconsistencies. Please refer to my technical comment #5 listed in the review report.
Author Response
1) In the text body of the manuscript, please use rectangular brackets for the literature references instead of the round ones, which are suitable only for the equations.
Reply: The brackets have been modified.
2) All figure links are erroneous, they are mentioned in the manuscript with the error message “Error! Reference source not found”.
Reply: The errors have been corrected.
3) Unfortunately, the article is written carelessly regarding the English language and the presentation of the applied methodology. The worst example of carelessness in methodology presentation is demonstrated in Section 2.5.2, where three types of the results normalization methods are described. The authors repeated equations (11, 12, 13) with the unreadable content on page 5, lines 218-228. That duplication must be removed and the rest part of Section 2.5.2 re-written.
Reply: The equations (11-13) have been corrected.
4) Page 6, line 231: Please correct the definition of MAER. Instead of “absolute error ratio”, it should be “mean absolute error ratio”.
Reply: The definition has been corrected.
5) There are many examples of noun-verb inconsistencies. The inconsistency is related to using plural and singular forms in the noun-verb connections. Singular subjects require singular verb forms, while plural subjects require plural verb forms. For instance, please correct the following noun-verb inconsistencies:
5a) Page 6, line 256: “TMSVPMI, TAVPMI, TFPMI and VFPMI of PGMGVCE is larger than that of model in (16)”. You have the plural nouns “TMSVPMI, TAVPMI, TFPMI and VFPMI of PGMGVCE” and the singular verb “is”.
5b) Page 9, lines 291-292: “This shows that Sigmoid normalization outperform the other two slightly.” The singular form of the noun “normalization” is inconsistent with the plural verb “outperform”.
5c) Page 11, lines 355-357: “The ability of our model, PGMGVCE, to generate images with authentic textures and its potential to offer safer and more precise diagnostics represent a significant advancement in medical imaging.” The singular subject “ability” is inconsistent with the plural form of the verb “represent”.
Reply: The errors have been corrected.
Reviewer 2 Report
Comments and Suggestions for Authors
Dear Editor,
I would like to express my deep thanks to you for allowing me to review this valuable manuscript, " Pixelwise Gradient Model with GAN for Virtual Contrast Enhancement in MRI Imaging". This paper provides a thorough analysis of different alterations of the PGMGVCE model. The comparisons involve changes in hyperparameters, utilization of various image-normalizing approaches, and exclusive training of the models with either T1-w or T2-w pictures.
The paper needs the following changes:
1. The authors need to expand on medical imaging in the introduction section
2. The abbreviation GAN should be defined and subsequently abbreviated. The term GAN is initially introduced in the abstract section without a prior definition but is subsequently defined later on.
3. The whole manuscript needs careful proofreading for minor spacing, syntax, and language corrections in some places.
4. A list of abbreviations should be added to the manuscript
5. Tables must be self-explanatory; thus, abbreviations must be defined in footnotes so that readers may examine them without having to go to the main text
6. The authors created beautiful figures, but the figures in the manuscript are not clear enough. The original or high-definition Figures should be provided
7. All Figures not cited in the manuscript
Author Response
- The authors need to expand on medical imaging in the introduction section
Reply: A paragraph on previous literature has been added.
Change in text: The medical field has recently spotlighted the advancement of DL in generating syn-thesized images.[17-23] Deep neural networks' ability to dissect and understand the intricate details of tumor characteristics has led to the innovation of creating VCE-MRI images from non-contrast MRI scans for brain cancer patients.[7, 24] Specifically, Gong et al. utilized a U-shaped DL model to merge MRI images without gadolinium-based contrast agent and with a low dose of gadolinium-based contrast agent, achieving VCE-MRI images that mimic those produced with a full dose of gadolinium-based contrast agent. This study showcased the potential of DL in extracting contrast enhancement details from full-dose gadolinium-based contrast agent MRI images and generating vceT1w MRI im-ages of satisfactory quality. Building on this groundwork, A three-dimensional Bayesian neural network that integrates ten different MRI techniques to produce VCE-MRI images was introduced[7]. This confirmed the DL network's capacity to utilize various non-contrast imaging methods for synthesizing images. Despite these encouraging results, the current DL models face challenges in harnessing the full potential of the diverse in-formation available from different imaging inputs. This limitation's impact becomes more significant when diagnosing deeply infiltrative NPC, due to the complex interaction of pixel intensity across different imaging modalities.[25]
- The abbreviation GAN should be defined and subsequently abbreviated. The term GAN is initially introduced in the abstract section without a prior definition but is subsequently defined later on.
Reply: “Our model, PGMGVCE, employs pixelwise gradient methods to delineate the shape of VCE images, complemented by the use of a Generative Adversarial Network (GAN) to replicate the image contrast of VCE.” Defines GAN
- The whole manuscript needs careful proofreading for minor spacing, syntax, and language corrections in some places.
Reply: The manuscript has been proofread again.
- A list of abbreviations should be added to the manuscript
Reply: Can you please provide sample manuscripts on the format of how to include abbreviations to text?
- Tablesmust be self-explanatory; thus, abbreviations must be defined in footnotes so that readers may examine them without having to go to the main text
Reply: The abbreviations have been supplied with the full terms.
- The authorscreated beautiful figures, but the figures in the manuscript are not clear enough. The original or high-definition Figures should be provided
Reply: All figures will be provided to the editor separately.
- All Figures not cited in the manuscript
Reply: The figures are cited but they are shown as “Error! Reference source not found”. They have been corrected.
Reviewer 3 Report
Comments and Suggestions for Authors
The article entitled “Pixelwise Gradient Model with GAN for Virtual Contrast Enhancement in MRI Imaging” is well-written and, from my point of view, would be of interest for the readers of Cancers. In spite of this,and before its publication, I would suggest authors to perform the following changes:
Line 32: substitute conclusion by conclusions.
Introduction: please give more details about the target of this research and, also, describe briefly the layout of the manuscript and not only sections 2.5 and 3.2.
Line 95 it is said Error! Reference source not found. Please correct it.
Please check how references must be cited in the text.
Formulae: please center them in the text and make sure that they are properly integrated in the manuscript.
Line 182 you refer to Research Ethics Committee in Hong Kong. Is it an ethics committee of any institution? Is it a committe of the city?
Lines 219-221: please check the text in these lines. What is it?
Line 246 it is said Error! Reference source not found. Please correct it.
Line 268 it is said Error! Reference source not found. Please correct it.
Line 283 it is said Error! Reference source not found. Please correct it.
Line 296 it is said Error! Reference source not found. Please correct it.
Author Response
The article entitled “Pixelwise Gradient Model with GAN for Virtual Contrast Enhancement in MRI Imaging” is well-written and, from my point of view, would be of interest for the readers of Cancers. In spite of this,and before its publication, I would suggest authors to perform the following changes:
Line 32: substitute conclusion by conclusions.
Reply: The error has been corrected.
Introduction: please give more details about the target of this research and, also, describe briefly the layout of the manuscript and not only sections 2.5 and 3.2.
Reply: Details of the novelty of the research and layout of the manuscript have been added.
Change in text: It was observed that the VCE images generated using the model in [22] exhibit excessively smooth textures compared to actual T1C images. The novelty of PGMGVCE, based on pixelwise gradient techniques, is that it demonstrates a texture in VCE images more akin to realistic T1C images.
Sections 2.1 and 2.2 introduce the model architectures of the PGMGVCE model. Section 2.3 introduces methods to evaluate the performance of the models. Section 2.4 discusses the data preprocessing steps. Section 3.1 shows the results of the VCE images. Sections 2.5 and 3.2 include comprehensive comparisons of various adaptations of the PGMGVCE model. Sections 4 and 5 are the discussion and conclusions, respectively.
Line 95 it is said Error! Reference source not found. Please correct it.
Reply: The error has been corrected.
Please check how references must be cited in the text.
Reply: The citation of references has been corrected.
Formulae: please center them in the text and make sure that they are properly integrated in the manuscript.
Reply: The formulae have been corrected/
Line 182 you refer to Research Ethics Committee in Hong Kong. Is it an ethics committee of any institution? Is it a committe of the city?
Reply: It is a ethics committee of the city.
Lines 219-221: please check the text in these lines. What is it?
Reply: The error has been corrected.
Line 246 it is said Error! Reference source not found. Please correct it.
Reply: The error has been corrected.
Line 268 it is said Error! Reference source not found. Please correct it.
Reply: The error has been corrected.
Line 283 it is said Error! Reference source not found. Please correct it.
Reply: The error has been corrected.
Line 296 it is said Error! Reference source not found. Please correct it.
Reply: The error has been corrected.
Reviewer 4 Report
Comments and Suggestions for Authors
The manuscript introduces a novel approach, the Pixelwise Gradient Model with GAN for Virtual Contrast Enhancement (PGMGVCE), for improving diagnostic accuracy in medical imaging, specifically in Magnetic Resonance Imaging (MRI) for nasopharyngeal cancer (NPC). PGMGVCE utilizes pixelwise gradient methods with Generative Adversarial Networks (GANs) to enhance T1-weighted (T1-w) and T2-weighted (T2-w) MRI images, simulating the effects of gadolinium-based contrast agents without associated risks. The model was optimized through variations, demonstrating similar accuracy to an existing model, but showing improvements in texture representation. PGMGVCE presents an innovative and safe approach to Virtual Contrast Enhancement in MRI, showcasing the power of deep learning in medical imaging. It contributes to provide a safer alternative to contrast agents, potentially improving diagnostic precision, especially in the context of nasopharyngeal cancer detection and characterization.
Globally, the manuscript is very well written and organized. However, there are some issues that should be addressed.
English needs to be reviewed; please refer to the attached commented PDF document where some of the needed corrections are highlighted. There are also some typos and typing errors.
Correct the missing references to figures and tables.
Avoid repeat the definition of acronyms/abbreviations (e.g., PGMGVCE, MAER, MSER, SSIMR).
Lines 148-157, equations 4-6. I recommend the use of "g" for "g"round truth and “s” for “s”ynthetic to denote the respective type of images. It is confusing to use “g” for synthetic…
Correct the sentence in lines 158-159.
Lines 174-175: Why these kernels? Is it because one acts like a horizontal "amplifier" and the other vertical...? The choice should be justified.
Correct the contents of lines 218 to 225.

Please refer to the comments above.
Author Response
English needs to be reviewed; please refer to the attached commented PDF document where some of the needed corrections are highlighted. There are also some typos and typing errors.
Correct the missing references to figures and tables.
Reply: The figures are cited but they are shown as “Error! Reference source not found”. They have been corrected.
Avoid repeat the definition of acronyms/abbreviations (e.g., PGMGVCE, MAER, MSER, SSIMR).
Changes in text:
Comparative analysis with the model in (22) reveals that, while basic accuracy metrics (MAE, MSE, SSIM) are comparable, PGMGVCE demonstrates superior texture representation.
This conclusion is supported by higher MAER, MSER, and SSIMR values when using single modalities.
Lines 148-157, equations 4-6. I recommend the use of "g" for "g"round truth and “s” for “s”ynthetic to denote the respective type of images. It is confusing to use “g” for synthetic…
Reply: The definitions of y and g have been exchanged.
Change in text: y(x) and g(x) denote the synthetic VCE T1C images and the ground truth, respectively. μ_y(x) , μ_g(x) and σ_y(x) , σ_g(x) are the means and variances of the synthetic images and the ground truth, whereas σ_y(x)g(x) is the covariance of y(x) and g(x).
Correct the sentence in lines 158-159.
Reply: The error has been corrected.
Change in text: To quantify the smoothness of VCE T1C images, four metrics are introduced. It can be used as a measure to check the difference in texture of the images.
Lines 174-175: Why these kernels? Is it because one acts like a horizontal "amplifier" and the other vertical...? The choice should be justified.
Reply: The kernels are chosen because they are the Sobel operators, which detect boundary of objects.
Correct the contents of lines 218 to 225.
Reply: The error has been corrected.
Change in text: Z-score normalization, or standard score normalization, involves rescaling the data to have a mean of 0 and a standard deviation of 1. Z-score normalization ensures that each feature contributes equally to the analysis, which is critical when combining features of different scales and units. It enhances the model's sensitivity to outliers, which can be vital for identifying anomalies in medical images. When the underlying data distribution is Gaussian, z-score normalization makes the features more Gaussian-like, which is an as-sumption in many machine learning models.
Sigmoid normalization transforms data using the sigmoid function to constrain val-ues within a range of 0 to 1. It bounds the input into a fixed range, which can be beneficial for models that are sensitive to input scale and distribution. The smooth nature of the sig-moid function provides smooth gradients, which can aid in the convergence during the training of deep learning models. In medical images, this can help preserve the context and relative contrast between different tissue types while standardizing the overall inten-sity scale.
Tanh normalization is similar to sigmoid but rescales the data to a range between 0 and 1. Data is centered around 1/2, which can lead to better performance in models where the sign of the data is important. The steeper gradient of tanh (compared to sigmoid) around the center can lead to faster learning and convergence in some cases. For medical images, this method can enhance contrast between areas of interest, potentially improving the model's ability to learn and distinguish pathological features.
Reviewer 5 Report
Comments and Suggestions for Authors
The authors of this paper introduce a novel approach for virtual contrast enhancement (VCE) in Magnetic Resonance Imaging (MRI), particularly focusing on nasopharyngeal cancer (NPC). The proposed model, Pixelwise Gradient Model with GAN for Virtual Contrast Enhancement (PGMGVCE), makes use of pixelwise gradient methods with Generative Adversarial Networks (GANs) to enhance T1-weighted (T1-w) and T2-17 weighted (T2-w) MRI images. This approach combines the benefits of both modalities to simulate the effects of gadolinium-based contrast agents, thereby reducing associated risks. Various modification of PGMGVCE, including changing hyperparameters, using normalization methods (z-score, Sigmoid, and Tanh) and training the model with T1-w or T2-w images only, were tested to optimize the model's performance. The paper is interesting. The writing is carefully developed and organised, but certain issues have been raised that require attention.
Introduction:
Please revise the reference style to follow the mdpi style.
Please highlight the novelty and contributions in the introduction.
Please add a section on related works and add previous relevant studies.
Method
Please correct the statement in line 95 “The PGMGVCE architecture is depicted in Error! Reference source not found.”
Please check the alignements of the equations.
Figures sre not cited in the text. They should be cited in the text.
Some equations are written in a very small font.
Please add sample images to the dataset description.
Experiments
Please check this statement in line 268: “Error! Reference source not found.”
Can you provide the number of epochs and additional information on hyperparameter fine-tuning? Furthermore, it is advisable to share your code to ensure reproducibility and facilitate comparison.
What are the limitations of the study
Please compare your results with other methods
The conclusion
please add your future directions.
Author Response
The authors of this paper introduce a novel approach for virtual contrast enhancement (VCE) in Magnetic Resonance Imaging (MRI), particularly focusing on nasopharyngeal cancer (NPC). The proposed model, Pixelwise Gradient Model with GAN for Virtual Contrast Enhancement (PGMGVCE), makes use of pixelwise gradient methods with Generative Adversarial Networks (GANs) to enhance T1-weighted (T1-w) and T2-17 weighted (T2-w) MRI images. This approach combines the benefits of both modalities to simulate the effects of gadolinium-based contrast agents, thereby reducing associated risks. Various modification of PGMGVCE, including changing hyperparameters, using normalization methods (z-score, Sigmoid, and Tanh) and training the model with T1-w or T2-w images only, were tested to optimize the model's performance. The paper is interesting. The writing is carefully developed and organised, but certain issues have been raised that require attention.
Introduction:
Please revise the reference style to follow the mdpi style.
Reply: The style of reference has been modified.
Please highlight the novelty and contributions in the introduction.
Reply: Novelty and contributions are highlighted
Change in text: Pixelwise gradient was originated from image registration [26-28] and, to our knowledge, our work is the first to apply to image synthesis, in particular, VCE-MRI. The evaluation of our models encompasses not only quantitative accuracy metrics, such as mean absolute error (MAE), mean square error (MSE), and structural similari-ty (SSIM), but also qualitative assessments of texture. It was observed that the VCE images generated using the model in [22] exhibit excessively smooth textures com-pared to actual T1C images. The novelty of PGMGVCE, based on pixelwise gradient techniques, is that it demonstrates a texture in VCE images more akin to realistic T1C images. This is evidenced by various metrics introduced in this paper, such as the Total Mean Square Variation per Mean Intensity (TMSVPMI), Total Absolute Variation per Mean Intensity (TAVPMI), Tenengrad Function per Mean Intensity (TFPMI), and Var-iance Function per Mean Intensity (VFPMI). Despite similar mean absolute errors be-tween images produced by PGMGVCE and the model in [22] when compared with ground truth, the improved textural fidelity of PGMGVCE images suggests its superi-ority over the model in [22].
Please add a section on related works and add previous relevant studies.
Change in text: The medical field has recently spotlighted the advancement of DL in generating synthesized images. [17-23] Deep neural networks' ability to dissect and understand the intricate details of tumor characteristics has led to the innovation of creating VCE-MRI images from non-contrast MRI scans for brain cancer patients. [7, 24] Spe-cifically, Gong et al. utilized a U-shaped DL model to merge MRI images without gado-linium-based contrast agent and with a low dose of gadolinium-based contrast agent, achieving VCE-MRI images that mimic those produced with a full dose of gadolini-um-based contrast agent. This study showcased the potential of DL in extracting con-trast enhancement details from full-dose gadolinium-based contrast agent MRI images and generating vceT1w MRI images of satisfactory quality. Building on this ground-work, A three-dimensional Bayesian neural network that integrates ten different MRI techniques to produce VCE-MRI images was introduced [7]. This confirmed the DL network's capacity to utilize various non-contrast imaging methods for synthesizing images. Despite these encouraging results, the current DL models face challenges in harnessing the full potential of the diverse information available from different imag-ing inputs. This limitation's impact becomes more significant when diagnosing deeply infiltrative NPC, due to the complex interaction of pixel intensity across different im-aging modalities. [25]
Method
Please correct the statement in line 95 “The PGMGVCE architecture is depicted in Error! Reference source not found.”
Reply: The error message has been corrected.
Please check the alignements of the equations.
Reply: The equations have been aligned.
Figures sre not cited in the text. They should be cited in the text.
Reply: The figures are cited but they are shown as “Error! Reference source not found”. They have been corrected.
Some equations are written in a very small font.
Please add sample images to the dataset description.
Reply: Figure 2 has been added as sample images
Experiments
Please check this statement in line 268: “Error! Reference source not found.”
Reply: The error has been corrected.
Can you provide the number of epochs and additional information on hyperparameter fine-tuning? Furthermore, it is advisable to share your code to ensure reproducibility and facilitate comparison.
Reply: For hyperparameter fine-tuning, the only hyperparameters in the model are the pixelwise gradient loss term and GAN loss term and their fine-tunings are described in section 3.2.1. The codes can be shared upon request.
Change in text: We performed 14000 iterations for training.
Data Availability Statement: For original data and computer programs, please contact the first author (K..H.C.)(khcheng9209@gmail.com).
What are the limitations of the study
Change in text: A limiting aspect of our synthesis network's efficacy is its training solely on T1- and T2-weighted MRI images. It appears these types of images may not encapsulate all neces-sary details for effective contrast synthesis. This issue could potentially be mitigated by incorporating additional MRI techniques (like diffusion-weighted MRI) into our network's input. Furthermore, the model has only undergone evaluation using a single data set. Therefore, its performance and the ability to generalize across different scenarios require further examination in subsequent research.
Please compare your results with other methods
Reply: Our model has been compared with the existing one in [22]. To our knowledge, this is the only study of VCE of NPC patients, so comparisons with previous methods are limited.
The conclusion
please add your future directions.
Change in text: One possible future direction would be performing segmentation on the tumor region to evaluate the performance of the tumor enhancement. Different VCE methods and the ground truth real T1C can be segmented to compare with each other. This would be a method to test if the tumor contrast of the VCE images is enhanced.
Round 2
Reviewer 3 Report
Comments and Suggestions for Authors
After the changes performed by the authors, I consider that the article is ready for its publication.
Reviewer 5 Report
Comments and Suggestions for Authors
The authors have addressed my comments